# Mechanochemically accessing a challenging-to-synthesize depolymerizable polymer

Tze-Gang Hsu[1,3], Shiqi Liu[1,3], Xin Guan[1], Seiyoung Yoon[1], Junfeng Zhou[1], Wei-Yuan Chen[2], Sanjay Gaire[2], Joshua Seylar[1], Hanlin Chen[1], Zeyu Wang [1], Jared Rivera[1], Leyao Wu[1], Christopher J. Ziegler[2], Ruel McKenzie[1] & Junpeng Wang [1] ✉

Polymers with low ceiling temperatures ($T_c$) are highly desirable as they can depolymerize under mild conditions, but they typically suffer from demanding synthetic conditions and poor stability. We envision that this challenge can be addressed by developing high-$T_c$ polymers that can be converted into low-$T_c$ polymers on demand. Here, we demonstrate the mechanochemical generation of a low-$T_c$ polymer, poly(2,5-dihydrofuran) (PDHF), from an unsaturated polyether that contains cyclobutane-fused THF in each repeat unit. Upon mechanically induced cycloreversion of cyclobutane, each repeat unit generates three repeat units of PDHF. The resulting PDHF completely depolymerizes into 2,5-dihydrofuran in the presence of a ruthenium catalyst. The mechanochemical generation of the otherwise difficult-to-synthesize PDHF highlights the power of polymer mechanochemistry in accessing elusive structures. The concept of mechanochemically regulating the $T_c$ of polymers can be applied to develop next-generation sustainable plastics.

Synthetic polymers are among the most important materials in modern society due to their light weight, low cost, and robustness. However, the durability of polymers has caused environmental pollution and sustainability issues. The stable nature of conventional synthetic polymers can be partly ascribed to their high ceiling temperatures ($T_c$), which is defined as the temperature where polymerization and depolymerization reach an equilibrium. On the other hand, low-$T_c$ polymers have metastable structures that are more susceptible to depolymerization. A low $T_c$ is especially useful for triggered degradation in polymers. For example, polyphthalaldehyde, a well-known low-$T_c$ (−40 °C) polymer, has been employed in photoresist and transient devices[1–5]. However, these materials are not competitive with industrially used polymers due to their metastable nature and are therefore not suitable for use as sustainable plastics. In addition, the conditions for synthesizing these polymers are typically demanding (e.g., low temperatures, ultrapure monomers, and a strict anhydrous and oxygen-free environment), and the synthesis often suffers from low yield and/or low molecular weight (Fig. 1a)[6–8].

Recently, our group and others have demonstrated the concept of locked degradability, aiming to address the stability issue of degradable polymers[9–12]. In the locked state, the polymer is nondegradable and highly stable. When recycling is needed, mechanochemical activation unlocks the degradability and converts the polymer into a degradable one. We envisioned that this strategy of locking and unlocking can be applied to address the stability issue of low $T_c$ polymers: if a high-$T_c$ polymer (locked state, for use and storage) can be converted into a low-$T_c$ polymer (unlocked state, for recycling), the paradox between stability and depolymerization can be broken (Fig. 1b).

Here, we demonstrate the concept of mechanochemically regulating $T_c$ with the 2,5-dihydrofuran (DHF) system, the ring-opening metathesis polymerization (ROMP) of which has proved challenging[13]. We designed a polymer **P1** (Fig. 1b), which is an unsaturated polyether containing cyclobutane-fused THF in each repeat unit. Upon force-induced cycloreversion of cyclobutane, each repeat unit is converted into three repeat units of poly(2,5-dihydrofuran) (PDHF). The resulting

[1]School of Polymer Science and Polymer Engineering, The University of Akron, 170 University Ave, Akron, OH 44325, USA. [2]Department of Chemistry, The University of Akron, 170 University Ave, Akron, OH 44325, USA. [3]These authors contributed equally: Tze-Gang Hsu, Shiqi Liu. ✉e-mail: jwang6@uakron.edu

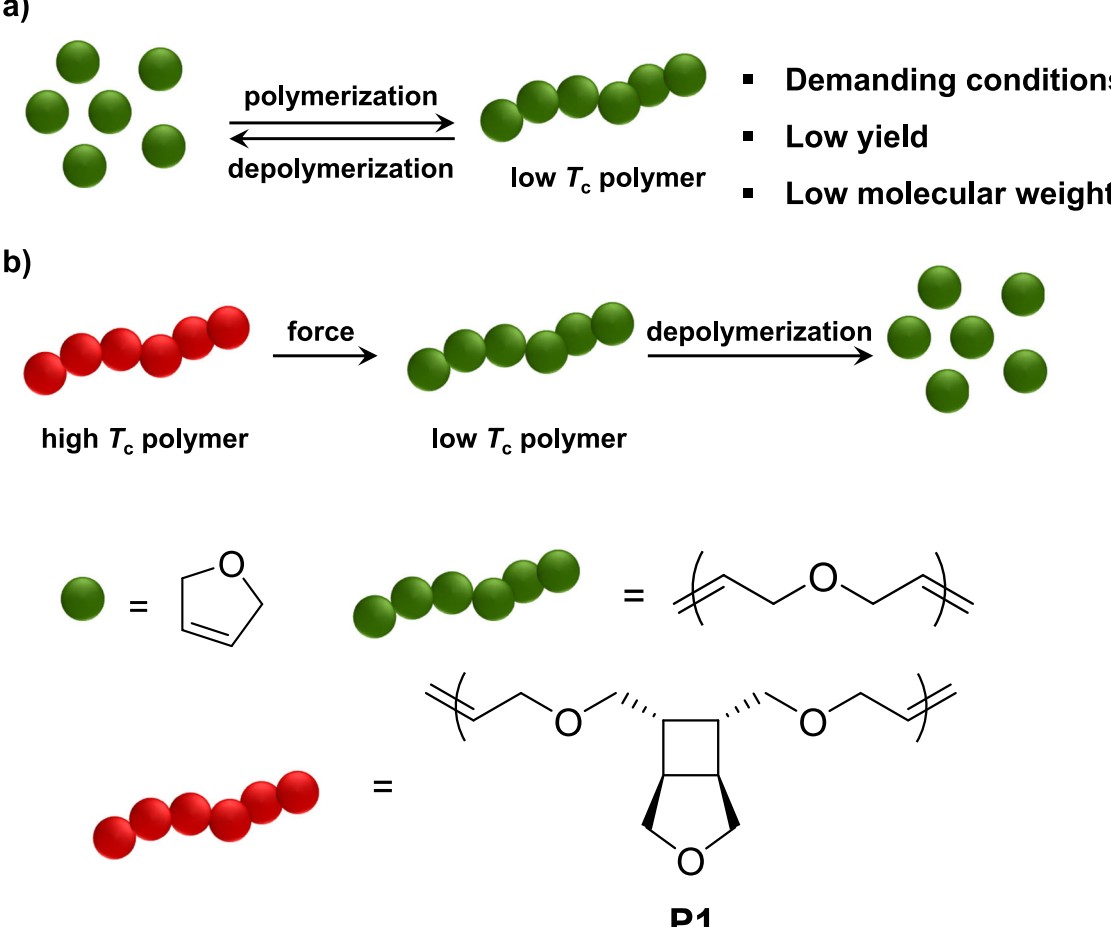

**Fig. 1 | Comparison between a polymer with low ceiling temperature ($T_c$) and this work. a** Polymerization of low-$T_c$ polymer requires demanding conditions and suffers from low yield and low molecular weight. **b** A high-$T_c$ polymer is mechanochemically converted into a low-$T_c$ polymer, which can undergo depolymerization. In this work, **P1** is the high-$T_c$ polymer, and poly(2,5-diydrofuran) is the low-$T_c$ polymer.

PDHF undergoes quantitative depolymerization to form DHF. The mechanochemical generation of PDHF and its catalytic depolymerization reveal the potential of applying polymer mechanochemistry to address challenges in plastics sustainability.

## Results and discussion

### Revisiting PDHF synthesis

As five-membered cyclic olefins (such as cyclopentene and 2,3-dihydrofuran) have relatively low ring strain energies (RSE) that enable reversible polymerization, they have been leveraged recently to develop chemically recyclable polymers[14,15]. While successes have been achieved with cyclopentene[6,14,16–18] and 2,3-dihydrofuran[15,19,20], the ROMP of DHF has been less prolific. For example, Höcker's attempts to polymerize DHF in bulk with a tungsten catalyst and with a chromium catalyst both failed[19]. Wagener reported the bulk ROMP of DHF using a molybdenum catalyst that yielded PDHF with a number-average molecular weight ($M_n$) >20 kDa, but the conversion was only 33%[13]. In addition, their acyclic diene metathesis polymerization using diallyl ether as the monomer afforded DHF and PDHF at conversions of 63% and 37%, respectively, and the latter had a $M_n$ of 640[13]. Compared to cyclopentene and 2,3-dihydrofuran, the difficulty in the ROMP of DHF can be attributed to its even lower RSE (Fig. 2a; RSEs were calculated according to a previously described procedure[17,21]).

To gain a better understanding of the reactivity and thermodynamics in the ROMP of DHF, we tested the polymerization in bulk using commercially available ruthenium catalysts. No polymer was observed when Grubbs 1st- and 2nd-generation catalysts (G1 and G2) were used (Supplementary Fig. 1 and Supplementary Fig. 2, respectively); a conversion of 23% was observed when G3 was used as the initiator, which yielded a PDHF with $M_n$ = 14 kDa (Fig. 2b). Our RSE calculations show that the 10-membered ring DHF dimer has a RSE of 7.0 kcal/mol (Fig. 2a), which is 3.6 kcal/mol higher than that of DHF, suggesting more favorable ROMP thermodynamics. However, the attempts to polymerize DHF dimer using G1, G2, and G3 all failed, and they only yielded DHF as the product (Fig. 2c). Note that because DHF dimer is a solid, solvent was added to dissolve it for polymerization, which diluted the concentration. It is likely that the formed PDHF depolymerized in situ to DHF due to the more dilute concentration. Taken together, these trials further highlight the low $T_c$ and proneness to depolymerization of PDHF due to the low RSE of DHF. The challenges in accessing PDHF motivated us to pursue a mechanochemical synthesis route.

### Design and synthesis of PDHF-generating mechanophore

In PDHF, every two adjacent alkenes are separated by a -CH$_2$-O-CH$_2$- spacer (Fig. 3a). Since the cycloreversion of cyclobutane generates two alkenes, the force-induced cycloreversion of a cyclobutane-locked -CH$_2$-O-CH$_2$- unit, (i.e., a cyclobutane-fused THF; Fig. 3a) will generate a structure in which the -CH$_2$-O-CH$_2$- spacer is connected by two alkenes (Fig. 3a). A reasonable design of the polymer that can generate PDHF upon mechanochemical activation is shown in Fig. 3b: Each repeat unit contains a cyclobutane-fused THF and two -CH$_2$-O-CH$_2$- spacers that

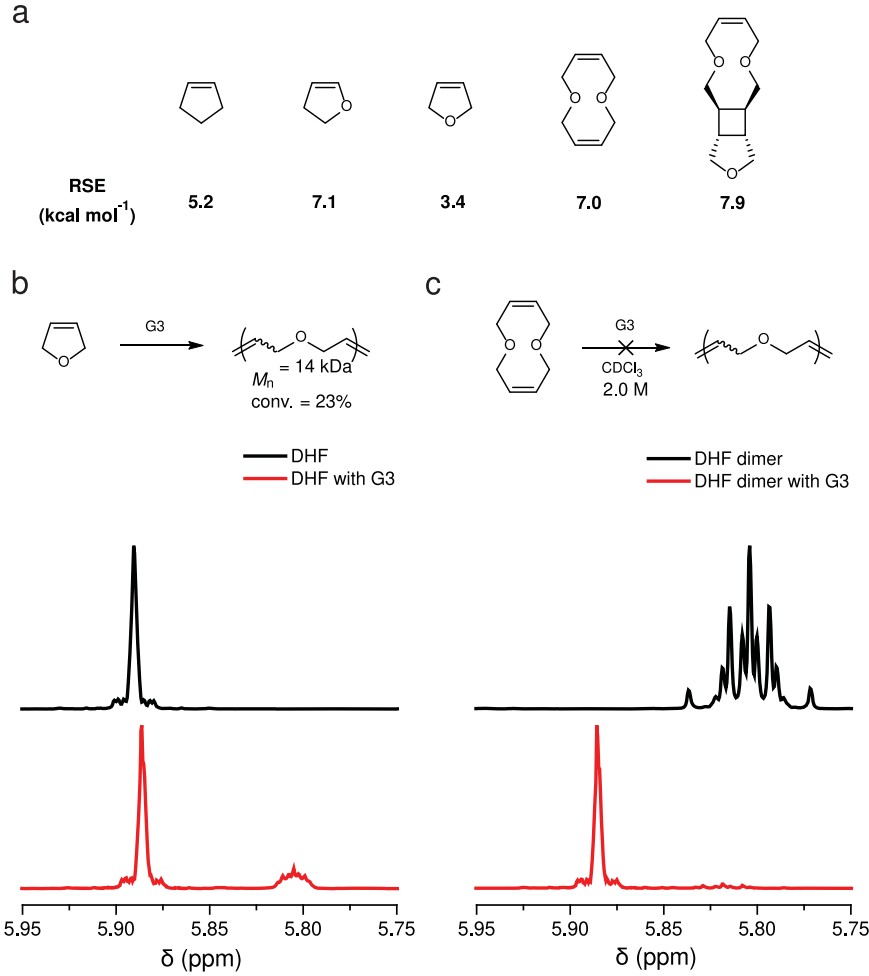

**Fig. 2 | Revisiting the synthesis of poly(2,5-dihydrofuran). a** Ring strain energies (RSE) of monomers relevant to 2,5-dihydrofuran (DHF). **b**, **c** Partial ${}^{1}$H NMR for DHF (**b**) and DHF dimer (**c**) before (in black) and after (in red) polymerization.

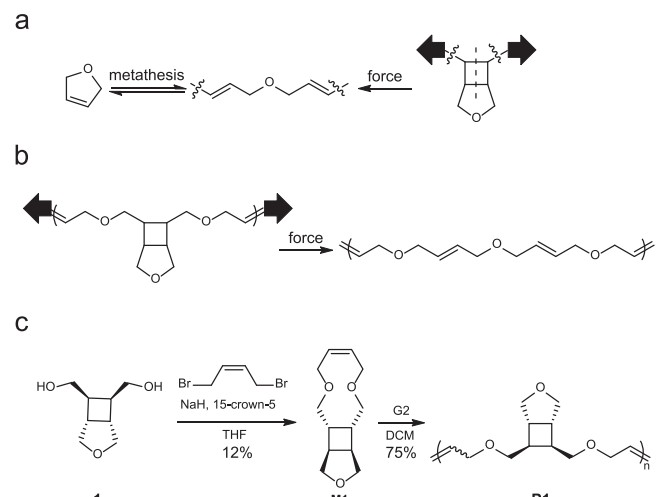

**Fig. 3 | Design and synthesis of the polymer that can mechanochemically generate PDHF. a** Reversible transformation between DHF and PDHF, and a mechanochemical pathway for a PDHF unit. **b** Structure of polymer that can mechanochemically generate PDHF. **c** Synthetic scheme of **M1** and **P1**.

connect the cyclobutane and an alkene; force-induced cycloreversion of each repeat unit would result in three repeat units of PDHF. This design of polymer corresponds to a 10-membered ring monomer, **M1**, which was conveniently synthesized through nucleophilic substitution

reactions between diol **1**[21] and *cis*−1,4-dibromo-2-butene (Fig. 3c). The use of 15-crown-5 ether was found to be essential for the successful cyclization, likely due to the increased nucleophilicity of the alkoxide as the crown ether sequesters sodium cation[22]. The structure of **M1** was confirmed by crystallography (Supplementary Fig. 34 and Supplementary Data 1). ROMP of **M1** using G2 afforded the corresponding polymer **P1** with a yield of 75%; different $M_n$ were obtained by varying the monomer-to-initiator ratio (Supplementary Table. 1). According to the ${}^{1}$H NMR spectrum (Fig. 4b), the *cis*- and *trans*-alkenes in **P1** have distinct proton peaks in both olefinic (5.9−5.6 ppm) and allylic (4.1−3.9 ppm) regions. Integration of the peaks show a *cis*-to-*trans* ratio of 1:5.3 (Supplementary Fig. 7). Thermogravimetric analysis of **P1** showed a decomposition onset temperature (the temperature at which 5% weight loss occurs) of 332 °C (Supplementary Fig. 4). The glass transition temperature ($T_g$) of **P1**, measured by differential scanning calorimetry, is −17 °C (Supplementary Fig. 4).

## Mechanochemical activation
To demonstrate the mechanochemical generation of PDHF, a THF solution of **P1** with a $M_n$ of 35 kDa was subjected to ultrasonication (power density: 9.26 W cm⁻²; temperature: 6–9 °C)[23]. Fig. 4b shows the ${}^{1}$H NMR spectra of **P1** before and after 2 h of sonication. Observation of new olefinic (e) and allylic (f) peaks in the sonicated polymer (**SP1**) supports the formation of PDHF. Mechanochemical activation of cyclobutane has been extensively studied, and it is well accepted that the cycloreversion proceeds through a stepwise pathway involving the formation of a diradical intermediate followed by stereochemistry-

determining alkene formation[9,11,12,24–27]. Analysis of the peaks revealed a mixture of *E* and *Z* alkenes in which more than 95% were *E* form (Fig. 4b), indicating that the initial stereochemistry is predominantly conserved. The retention in stereochemistry is in drastic contrast to previous sonication studies of ester-linked cyclobutane, where significant levels of inversion in the stereochemistry were observed[11,24,25]. The difference can be partly attributed to the different stability of the diradical intermediates between the alkyl- and ester-linked cyclobutanes[27]. According to a 2021 study by Liu et al. on the mechanochemical cycloreversion of cyclobutane-fused norbornene, the more stable ester-linked intermediate has greater susceptibility to relaxation, allowing for statistical dynamics to govern the product stereochemistry[27]. In their study, both *cis*- and *trans*-alkyl-substituted cyclobutane showed over 95% retention in stereochemistry. Nevertheless, in our recent study on a *trans*-cyclobutane-fused lactone, only 33% stereoretentive product was observed, despite the alkyl linker on cyclobutane[9]. To eliminate the effect from having different rings fused

to cyclobutane (THF vs. lactone), we synthesized a polymer **P2** that contains *trans*-cyclobutane-fused THF in its repeat unit (for details, see synthetic section of Supplementary Information). Sonication of **P2** under otherwise identical conditions showed significant inversion in stereochemistry (>38%, Supplementary Table. 2), implying the impact of substituent stereochemistry on the stereochemistry of the ring-opening products. The different trends in stereochemistry retention/inversion observed between cyclobutane-fused norbornene and our system can be attributed to the different rings connected to cyclobutane, which is consistent with our previous observation that functional groups connected to C3 and C4 (Fig. 4a) can also impact the stereochemistry of the cycloreversion products[9].

To evaluate the mechanochemical reactivity of the *cis*-cyclobutane-fused THF, a kinetic study was performed on the sonication of **P1**. Polymers of two different initial $M_n$, 89 kDa and 58 kDa, were subjected to sonication. During sonication, aliquots were taken periodically from the reaction solution for [1]H NMR and GPC analyses to monitor the process of mechanochemical activation. A reduction in molecular weight was observed with increased sonication time (Fig. 5a). At 90 min, both the 89 kDa and 58 kDa polymers reached $M_n$ ~40 kDa. Meanwhile, the percentage of cyclobutane ring opening increased over time, and after 90 min of sonication, **P1** with initial $M_n$ of 89 kDa and 58 kDa reached ring-opening percentages of 53% and 38%, respectively (Fig. 5b). Here, the greater extent of ring opening in the 89 kDa polymer can be ascribed to a higher accessible force for the long polymer chain. In addition, for the similar initial $M_n$ (~89 kDa), the ring opening in **P1** occurred faster than those for the *trans*-cyclobutane-fused THF polymer **P2** (Supplementary Table. 2) and the previously reported *trans*-cyclobutane-fused lactone polymer (Supplementary Fig. 12). The preferential reactivity of *cis*-cyclobutane over its *trans* analog is consistent with what was observed in other cyclobutane-based mechanophores and can be attributed to more efficient coupling of the force to the mechanophore in the *cis* configuration[24,28–30].

A typical control experiment to validate mechanochemical activation is to sonicate a polymer with a molecular weight below the limiting $M_n$[31]. Our recent study suggests that the limiting $M_n$ for the cyclobutane mechanophore under similar experimental conditions is ~10 kDa[9]. A **P1** of 5 kDa was prepared and sonicated for 30 min. The [1]H NMR spectrum indicated no reaction had occurred (Supplementary Fig. 10), and the polymer retention time remained unchanged (Supplementary Fig. 11), supporting the mechanical nature of the generation of PDHF. In addition, a 154 kDa **P1** was subjected to sonication for

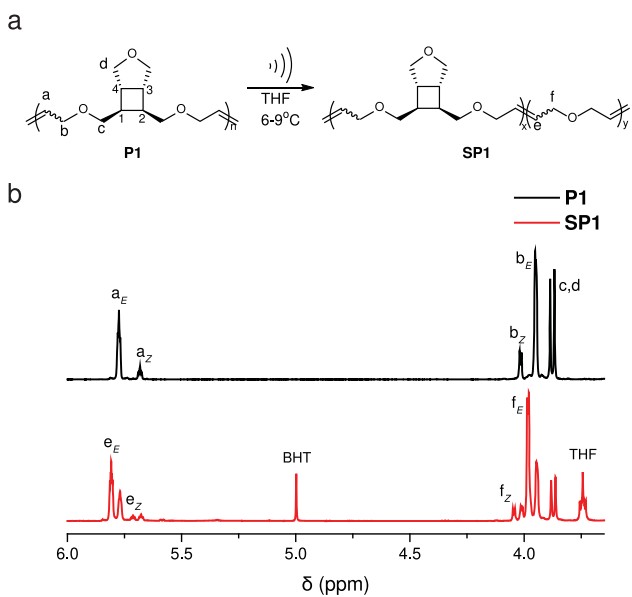

**Fig. 4 | Mechanochemical activation of** P1. **a** P1 is subjected to ultrasonication, resulting in **SP1**, which is a copolymer of **P1** and poly(2,5-diydrofuran). **b** Partial [1]H NMR of **P1** before (black) and after (red) sonication for 2 h.

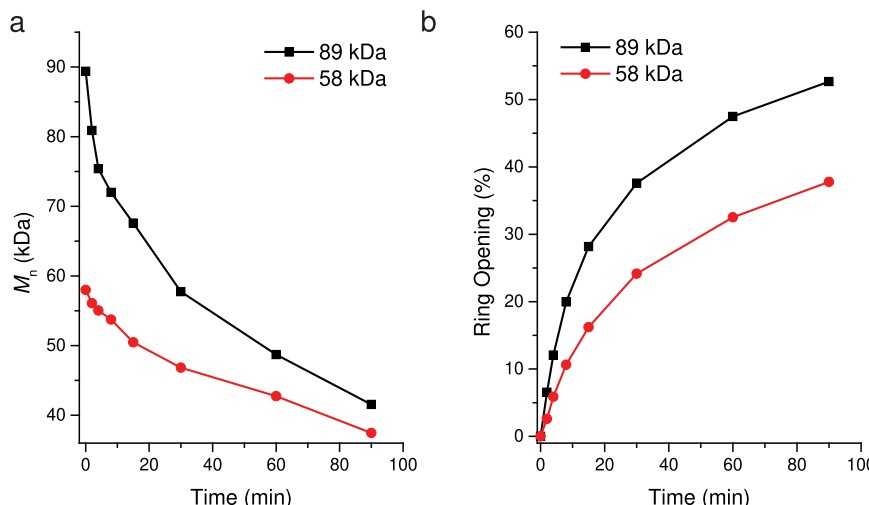

**Fig. 5 | Kinetic study of sonication.** Number-average molecular weight ($M_n$) (**a**) and percentage of ring opening (**b**) as a function of sonication time for **P1** with two different initial molecular weights (89 kDa, in black; 58 kDa, in red).

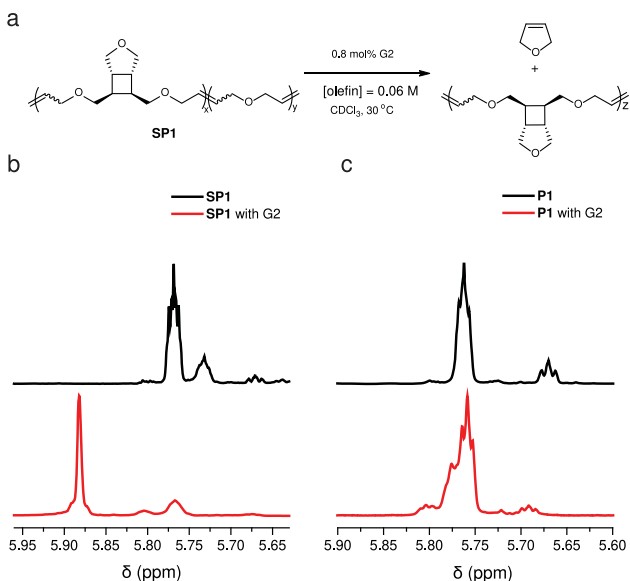

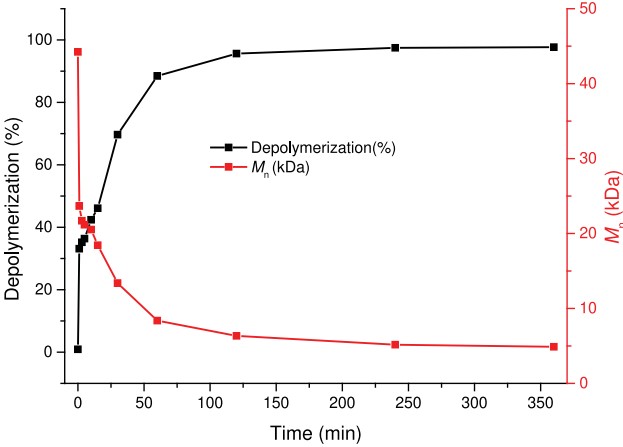

**Fig. 7 | Kinetic studies of SP1 depolymerization.** Percentage of depolymerization (in black) and number-average molecular weight of the residual polymer (in red) as a function of time.

**Fig. 6 | Depolymerization of SP1. a** The scheme for depolymerization of **SP1**. **b**, **c** Partial [1]H NMR spectra for **SP1** and **P1** before (black) and after (red) being stirred with G2 catalyst at 30 °C for 6 h.

1 h, resulting in an **SP1** with 70 kDa $M_n$ and 54% PDHF. Notably, the size of PDHF in the **SP1** is 38 kDa, which has not been achieved by the ROMP of DHF. This serves as another example of using polymer mechanochemistry to access chemical structures that are otherwise challenging to make[32,33].

## Depolymerization studies

**SP1** of 44 kDa with 68% PDHF (generated from sonication of 154 kDa for 4 h), was dissolved in CDCl$_3$ at a concentration of [olefin] = 0.06 M, and 0.8 mol% G2 was added. The reaction was allowed to proceed under 30 °C for 6 h (Fig. 6a). [1]H NMR of the resulting solution clearly showed olefinic peak of DHF in the range of 5.92–5.86 ppm (Fig. 6b), and integration of the peak correspond to a 98% conversion of PDHF to DHF (Supplementary Table. 3). The efficient depolymerization of PDHF also implies that **SP1** is mostly a block copolymer of **P1** and PDHF, similar to what was observed in sonication studies of other multi-mechanophore polymers[9–12,32,34]. In contrast, when **P1** was subjected to a similar condition ([olefin] = 0.025 M; 2 mol% G2), no formation of **M1** was observed from the [1]H NMR spectrum (Fig. 6c), indicating a relatively high $T_c$; the mass spectrometry analysis of the products showed a mixture of cyclic dimer, trimer, and tetramer of **M1** (Supplementary Fig. 18).

A kinetic study on the depolymerization of **SP1** was performed by taking aliquots at different time points for [1]H NMR and GPC characterization. At 5 min, depolymerization of the PDHF segment reached 33% (Fig. 7), and the depolymerization was complete at 2 h. Compared to previous studies on polypentenamer by Tuba et al., which showed complete depolymerization within 4 min at ambient temperature[35], the slower depolymerization kinetics of PDHF could be due to the weak coordination of the oxygen atoms (on the polyether and on the generated DHF) to the ruthenium catalyst center, an effect that has been observed previously in ruthenium-catalyzed olefin metathesis when solvents such as THF and diethyl ether were used[36]. In addition, GPC traces were found to gradually shift to a longer retention time (Supplementary Fig. 16), and $M_n$ of the residual polymer decreased to 6.3 kDa at 2 h of depolymerization, at which point the extent of PDHF depolymerization reached 96%. From 2 h to 6 h, although PDHF block had been mostly consumed, the molecular weight of the residual polymer continued decreasing to 4.9 kDa; the further reduction in molecular weight could be due to the intramolecular secondary metathesis on the **P1** block.

## Bulk mechanochemical activation

While solution-phase ultrasonication is suitable for testing the reactivity of mechanophores on a laboratory scale, the use of solvents diminishes its practicality as a technique for polymer degradation and recycling. In order to make the mechanically active degradable/depolymerizable polymers a real consideration as next-generation sustainable materials, activating mechanophores in bulk materials is essential. To test the feasibility of activating the cyclobutane-fused THF mechanophore in bulk, we first conducted an extrusion experiment on **P1**. 200 mg of **P1** ($M_n$ = 69 kDa, Đ = 1.80) was blended with 20 g of polyethylene glycol (PEG, $M_n$ ~ 2 kDa), and the blend was subjected to extrusion on a twin screw extruder with the processing temperature set at 70 °C. The extrusion product was precipitated in water to remove PEG. [1]H NMR of the resulting product showed minimal change compared to the original polymer (Fig. 8), indicating an insignificant extent of mechanophore activation. GPC trace of the extruded product showed a reduction in molecular weight (Supplementary Fig. 22), likely due to thermomechanical degradation, which has been observed during the extrusion of polyolefins[37]. The collected sample was then stirred with G2 in CDCl$_3$ overnight, and [1]H NMR of the resulting product showed an olefinic peak of DHF in the range of 5.92–5.86 ppm. Integration of the peaks showed 3% DHF relative to the total olefins, corresponding to 1% ring opening of the cyclobutane-fused THF mechanophore, considering that the activation of each mechanophore generates three PDHF units (Fig. 3b). GPC trace of the G2-treated sample showed no polymer peak left (Supplementary Fig. 22), indicating that the polymer was converted into oligomers through secondary metathesis.

To further explore the activation of the cyclobutane-fused THF mechanophore in bulk materials, the mechanophore was incorporated into a polymer network **PN1** by copolymerizing **M1** with a bis-cyclooctene crosslinker (Fig. 9a), which gave a transparent film (Fig. 9b). The film **PN1** was subjected to uniaxial compression testing, which showed Young's modulus of 1.03 ± 0.19 MPa. The film was compressed until 90% compressive strain, and the resulting material and 2 mol% G2 catalyst were added to CDCl$_3$, and the mixture was stirred at 30 °C for 16 h. The sol fraction of the resulting mixture showed no DHF formation on [1]H NMR (Fig. 9c).

While ball milling has been extensively applied to mechanochemically degrade commercial polymers[38–42], such as polystyrene, poly (ethylene terephthalate), and epoxy resin, its use in the context of

mechanophore activation is rare. To test the feasibility of activating the mechanophore through ball mill grinding, the film **PN1** was subjected to ball milling (25 Hz, 10 min) followed by depolymerization (2 mol% G2, CDCl$_3$, 30 °C, 16 h). $^1$H NMR of the sol fraction showed a

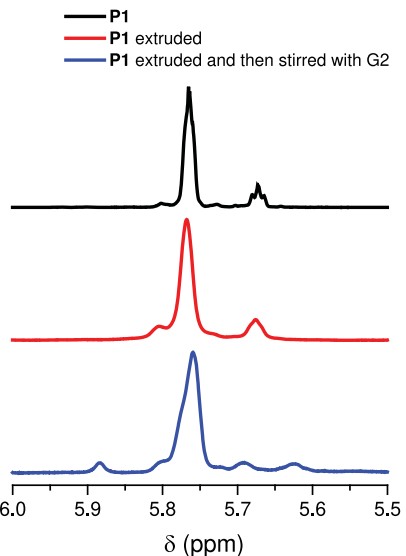

**Fig. 8 | Extrusion and depolymerization of** P1. Partial $^1$HNMR spectra for **P1** (top, black), extruded **P1** (middle, red), and extruded **P1** stirred with G2 overnight (bottom, blue).

pronounced peak in the range of 5.92–5.86 ppm, indicating the formation of a significant amount of DHF (Fig. 9c). Integration of the peaks showed 6 mol% DHF compared to the original olefin in **M1** (Supplementary Fig. 23). To quantify the extent of mechanophore activation, the ball milled film (prior to depolymerization) was subjected to hydrolysis in 0.1 M KOH solution in H$_2$O/DMSO (5/95) to degrade the ester group in the crosslinker. The de-crosslinked linear polymer sample was extracted with CDCl$_3$ and characterized with $^1$H NMR. As shown in Fig. 9d, both olefinic and allylic protons of PDHF were observed, and integration of the peaks showed 17% of mechanophore activation (Supplementary Fig. 26.). The 17% mechanophore activation should give ~40% DHF generation if all the PDHF units generated were converted into DHF. The substantially lower (only 6%) fraction of DHF generation could be due to the presence of the non-depolymerizable polyoctenamer units and the statistical (instead of blocky) distribution of PDHF along the polymer backbone. Nevertheless, the significant level of mechanophore activation achieved here sets a promising direction of mechanically activating and degrading this class of polymers in bulk using ball milling.

We have demonstrated the mechanochemical generation of a low-$T_c$ polymer PDHF that can depolymerize into DHF in the presence of a ruthenium catalyst. The mechanochemical activation enabled depolymerization demonstrated here can enrich the growing toolbox for force-induced small molecule release[43–46]. The utility of cyclobutane in controlling depolymerization of polymers again captures the versatility of this mechanophore, as it has been used to toughen hydrogels[47], access conjugated polymers[32,33], regulate reactivity[25], and control polymer degradation[9–12]. In previous works on mechanically active degradable polymers, cyclobutane

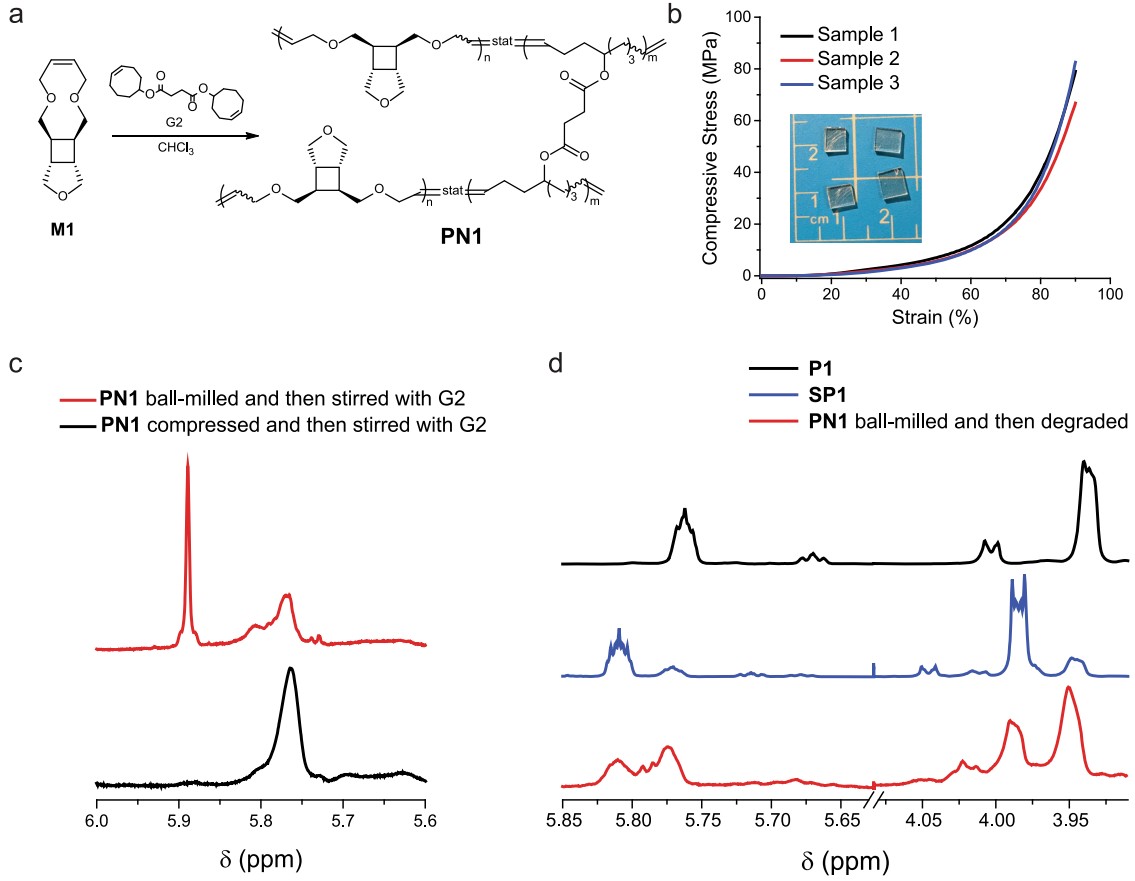

**Fig. 9 | Mechanochemical activation in polymer network** PN1. **a** Synthetic scheme of **PN1**. **b** Picture of **PN1** thin film and the stress–strain curve obtained from compressive testing of the film (sample 1, black; sample 2, red; sample 3, blue). **c** Partial $^1$H NMR spectra for ball milled (top, red) and compressed (bottom, black) **PN1** after being stirred with G2 catalyst overnight. **d** Partial $^1$H NMR spectra for **P1** (top, black), **SP1** (middle, blue), and hydrolyzed **PN1** (bottom, red).

was used to lock the degradable functionalities[9–12]; here, the function of cyclobutane is distinct in that the two alkenes generated from the cycloreversion of cyclobutane are used for olefin metathesis. In addition to the generation of olefins, cyclobutane also regulates the number of spacer atoms between the generated alkenes, which determines the size of the corresponding monomer, and in doing so, adjusts the $T_c$ of the resulting polymer. Since the connection between the ring size of monomers and the $T_c$ of polymers generally exists in all ring-opening polymerization systems, the strategy of mechanochemically regulating the number of spacer atoms and $T_c$ of polymers can be applied to other types of chemically recyclable polymers[48,49].

## Methods

### Materials

Tetrahydrofuran (THF), $N,N$-dimethylformamide (DMF) were obtained from Sigma-Aldrich and dried by immersing in 4 Å molecular sieves desiccant (beads, 4–8 mesh, Sigma-Aldrich) overnight before use. $CDCl_3$ and DMSO-d6 were purchased from Cambridge Isotope Laboratories. Other solvents were purchased from Fisher Chemical and used without further purification. All other chemicals were purchased from Sigma-Aldrich or Fisher Chemical without further purification before use unless specified.

### Instruments

All GPC experiments were carried out using Tosoh EcoSEC HLC-8320 GPC with two 17393 TSKgel columns (7.8 mm ID × 30 cm, 13 μm) and one 17367-TSKgel Guard Column (7.5 mm ID × 7.5 cm, 13 μm) using preservative-free HPLC grade THF (obtained from Fisher Chemical) at a flow rate of 1 mL min⁻¹ at 40 °C. Purification by preparative GPC were done by LaboACE LC-5060 which was connected to two JAIGEL-2HR columns using HPLC grade chloroform with 0.75% ethanol as preservative (obtained from Fisher Chemical) at a flow rate of 10 mL min⁻¹. ¹H and ¹³C NMR spectra were collected on a Varian 500 MHz spectrometer using $CDCl_3$ or DMSO-d6 as the solvent and referenced to residual solvent peak (δ = 7.26 in ¹H, 77.16 in ¹³C for $CDCl_3$; δ = 2.5 in ¹H, 39.52 in ¹³C for DMSO-d6). X-ray crystallographic study was carried out using Bruker ApexII Duo with comounted Mo and Cu microfocus radiation sources and Quazar optics for increased source brightness. ESI-MS spectra were recorded on a Waters Synapt HDMS quadrupole/time-of-flight (Q-ToF) mass spectrometer (Waters, Beverly, MA) in positive ion mode. Samples were prepared to a final concentration of 1 μg mL⁻¹ in methanol. Each sample was introduced to the electrospray source via direct infusion at a flow rate of 5 μL/min, with source parameters as follows: capillary voltage: 3.15 kV; cone voltage: 35 V; sampling cone voltage: 3.2 V; source temperature: 90 °C; desolvation temperature: 150 °C. High resolution-atmospheric pressure chemical ionization mass spectra were acquired using a Waters Synapt HDMS Q-ToF Mass Spectrometer (Waters, Beverly, MA) equipped with an atmospheric solids analyses probe in positive ion mode.

### Calculations of RSEs

The RSEs of the cyclic olefins were calculated by following a method demonstrated by Grubbs and coworkers[17]. This method involves the calculation of the enthalpy change for the ring-closing metathesis of the acyclic diene that forms the cyclic olefin and ethylene. The ring-closing metathesis is an isodesmic reaction—the numbers and types of bonds of the reactant are identical to those of the products. Since there is no ring strain in the acyclic diene reactant or the ethylene products, the enthalpy change of the reaction should originate in the ring strain of the cyclic olefin (Eq. (1)).

$$\text{RSE} = \triangle H = H_{\text{ethene}} + H_{\text{cyclic olefin}} - H_{\text{diene}} \qquad (1)$$

$H_{\text{ethene}}$, $H_{\text{cyclic olefin}}$, and $H_{\text{diene}}$ are the calculated enthalpies at 298.15 K and 1.0 atm for ethene, the cyclic olefin, and the diene, respectively. The structures were first optimized with the Universal Force Field as implemented in the auto-optimization tool in Avogadro, followed by a conformer search with Avogadro. The optimized conformation was subjected to geometry optimization and vibrational frequency calculations at the level of B3LYP/6-31 G(d,p) using the Gaussian 16 revision C.01 software.

### Polymerization

**P1** of different molecular weights were made by ROMP of **M1**. The following is an example of the procedure: To a vial was added **M1** (100 mg, 0.48 mmol, 1 equiv.) and G2 (0.4 mg, 0.000476 mmol, 0.001 equiv., in 480 μL of DCM stock solution). The reaction was gently stirred for 3 h, and excess amount of ethyl vinyl ether (150 μL) was added and stirred for 30 min to terminate the polymerization. The solvent was then removed on a rotavaper and subjected to prepGPC for further purification to yield transparent viscous oil, which was dried under vacuum overnight to afford 75 mg of polymer **P1**. (75 mg, yield: 75%, $M_n$ = 154 kDa, Đ = 1.97)

### Sonication

Sonication experiments were conducted using a Vibracell model VCX500 (Sonics & Materials) with a standard solid probe (tip diameter = 13 mm, titanium alloy Ti-6Al-4V) operated at 20 kHz. 1 mg/1 mL THF solution of **P1** was made and transferred to a reaction vessel (made by U Akron Glassblowing Shop) and was then deoxygenated by bubbling through $N_2$ for 30 min. The solution was sonicated under $N_2$ using a pulse sequence 1 s on/1 s off at an energy density of 9.3 W/cm² (AMPL = 25%). The temperature of the solution was maintained by placing the vessel in an ice/water bath. Ring-opening percentage and molecular weight changes were tracked by ¹H NMR and GPC, respectively.

### Depolymerization

The sonicated sample **SP1** (33 mg, %RO = 68%) was dried and reacted with 0.8 mol% of G2 (2.7 mg, 0.0031 mmol, 0.008 equiv., in 6.3 mL of degassed $CDCl_3$ stock solution) to olefin (0.06 M, 1.0 equiv) under the protection of $N_2$ for 6 h at 30 °C. Aliquots were taken out at different reaction periods, and excess amount of ethyl vinyl ether (100 μL) was added and left standing for 30 min before ¹HNMR and GPC analysis.

## Data availability

Crystallographic data for the structures in this Article have been deposited at the Cambridge Crystallographic Data Centre (CCDC) under deposition no. 2179611 (**M1**). All other data supporting the findings of this study are available within the Article and its Supplementary Information. Source data are provided with this paper.

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

## Acknowledgements

This material is based on work supported by the University of Akron and the National Science Foundation under grant no. CHE-2204079. T.-G.H. acknowledges Sigma Xi Grants-in-Aid of Research (G03152021120132966) for financial support. The authors thank Kayla Williams Pavlantos and Prof. Chrys Wesdemiotis for mass spectrometry analysis. The single-crystal structures were characterized with an X-ray diffractometer supported by the National Science Foundation (CHE-0840446 to C.J.Z.). The computational resources were provided by Extreme Science and Engineering Discovery Environment (TG-CHE220003). The authors thank The Ohio Board of Regents and The National Science Foundation (CHE-0341701 and DMR-0414599) for the funds used to purchase the NMR instrument used in this work.

## Author contributions

J.W. conceived the project and directed the research. H.C. and J.W. performed the density functional theory calculations and analyzed the computational data. T.-G.H., S.L., G.X., J.Z., J.R., J.S., and R.M. conducted the monomer and polymer syntheses. W.-Y.C., S.G., and C.J.Z. collected and analyzed the single-crystal data. S.Y., Z.W., and T.-G.H. conducted the thermal characterization of the polymers. T.-G.H. conducted solution-phase polymer mechanochemical activation and depolymerization. Z.W., L.W. S.Y., and T.-G.H. conducted bulk mechanochemical activation and depolymerization studies. T.-G.H. and J.W. prepared the manuscript.

## Competing interests

The authors declare no competing interests.
