## [Peer Review File · Nature Communications]

Mechanochemically accessing a challenging-to-synthesize depolymerizable polymerReviewers' Comments:

Reviewer #1:

Remarks to the Author:

Wang and coworkers report on a poly(2,5-dihydrofuran) with included cyclobutane mechanophores that gates the degradability by depolymerization through switching the ceiling temperature of the polymer. Frankly, the polymer that has been used/generated is commercially irrelevant, the generality and transferability of the proposed reaction mechanism is questionable, and the necessity for the addition of a depolymerization catalyst is cumbersome and a formidable "gate" in itself. BUT: Switching/manipulating the ceiling temperature of polymers is a major approach for a circular economy and not only degrading plastic waste but also reobtaining the polymerizable monomers. This holds true also for polymer mechanochemistry, which I anticipate to be a main research ground in sustainable polymer science in the near future. Hence, the demonstration to mechanochemically change the ceiling temperature of a polymer as a concept is a major key achievement in this direction. Obviously, there's a small trick involved as the monomers that are received are not the ones used for polymerization so no complete circularity was demonstrated. However, the principle still stands and I believe that this work is a very important contribution furthering future endeavors of many other researchers in the field.

The manuscript is well written and I cannot identify any scientific problems with it. All pertinent references to polymer mechanochemical manuscripts dealing with the force-induced depolymerization of polymers are included. I suggest, to broaden the scope, to pick up some of the literature dealing with force-induced mechanophore-free depolymerization reactions in ball mills though (e.g., 10.1021/acssuschemeng.1c03221, 10.1002/app.49628, 10.1021/acs.macromol.0c01510, or 10.1021/acs.molpharmaceut.0c00376). Trituration and polymer mechanochemistry rarely connect, but I believe there will be many synergies on this specific topic in the future.

To make this manuscript future-proof I suggest running a proof-of-concept experiment and see whether extruding this polymer in a commercial twin-screw extruder would also work for this purpose. Frankly, ultrasonication delivers extremely high shear rates but solvent and energy consumption will never make this method applicable on a commercial scale. I am aware that extruders require large sample amounts so maybe, if not available, the authors could blend a small fraction in another commercial polyolefin that does not interfere with the NMR or simply dialyze the non-reactive polymer after the extrusion to receive only the chain fragments. This would hold this manuscript to the same standard as reference 10 of the authors, by Craig and coworkers, also published in Nature Communications.

Otherwise, and despite the limitations of the usefulness of the investigated polymer, I believe this manuscript to be a milestone for the field and hence very suitable to be published in Nature Communications.

Reviewer #2:

Remarks to the Author:

In the present paper, Hsu et al. depict a new strategy to attain poly(2,5-dihydrofuran) mechanochemically from an unsaturated polyether cyclic alkene with a cyclobutene-fused as the mechanically active handle. This paper is extremely well written and well thought out. It builds upon some of the state-of-the-art on locked degradability.

This reviewer has very little to argue from a technical perspective. Where the paper falls a bit short of the lofty goals it sets out from the onset is the usefulness of such material. As such, the authors do not characterize either the processability (which may trigger activation) or the mechanical properties of the materials they synthesize? Would the ring opening occur under extrusion conditions or under load? While intellectually stimulating, that would seriously hamper the usefulness of the proposed materials. Addressing this point and matching the objectives with the data is in this reviewer's opinion critical.

We thank the reviewers for their thoughtful comments and suggestions. Responses can be found below in blue text.

Reviewer(s)' Comments to Author:

Reviewer: 1

Comments:

Wang and coworkers report on a poly(2,5-dihydrofuran) with included cyclobutane mechanophores that gates the degradability by depolymerization through switching the ceiling temperature of the polymer. Frankly, the polymer that has been used/generated is commercially irrelevant, the generality and transferability of the proposed reaction mechanism is questionable, and the necessity for the addition of a depolymerization catalyst is cumbersome and a formidable “gate” in itself. BUT: Switching/manipulating the ceiling temperature of polymers is a major approach for a circular economy and not only degrading plastic waste but also reobtaining the polymerizable monomers. This holds true also for polymer mechanochemistry, which I anticipate to be a main research ground in sustainable polymer science in the near future. Hence, the demonstration to mechanochemically change the ceiling temperature of a polymer as a concept is a major key achievement in this direction. Obviously, there's a small trick involved as the monomers that are received are not the ones used for polymerization so no complete circularity was demonstrated. However, the principle still stands and I believe that this work is a very important contribution furthering future endeavors of many other researchers in the field.

1. The manuscript is well written, and I cannot identify any scientific problems with it. All pertinent references to polymer mechanochemical manuscripts dealing with the force-induced depolymerization of polymers are included. I suggest, to broaden the scope, to pick up some of the literature dealing with force-induced mechanophore-free depolymerization reactions in ball mills though (e.g., 10.1021/acssuschemeng.1c03221, 10.1002/app.49628, 10.1021/acs.macromol.0c01510, or 10.1021/acs.molpharmaceut.0c00376). Trituration and polymer mechanochemistry rarely connect, but I believe there will be many synergies on this specific topic in the future.

We thank the reviewer for the suggested references. Force-induced mechanophore-free depolymerization by ball milling is indeed relevant to this work, and the references have been added to the revised manuscript. Motivated by the reviewer's comment, we tested the feasibility of mechanochemical activation using ball milling and obtained encouraging results—17% mechanophore activation after 10 min of ball mill grinding.

2. To make this manuscript future-proof I suggest running a proof-of-concept experiment and see whether extruding this polymer in a commercial twin-screw extruder would also work for this purpose. Frankly, ultrasonication delivers extremely high shear rates but solvent and energy consumption will never make this method applicable on a commercial scale. I am aware that extruders require large sample amounts so maybe, if not available, the authors could blend a small fraction in another commercial polyolefin that does not interfere with the NMR or simply dialyze the non-reactive polymer after the extrusion to receive only the chain fragments. This would hold this manuscript to the same standard as reference 10 of the authors, by Craig and coworkers, also published in Nature Communications.

We appreciate the reviewer's suggestion. As per the suggestion, we conducted extrusion on a blend of **P1**/PEG. In addition, we have also prepared a polymer network **PN1** using **M1** and a bis-cyclooctene crosslinker and performed compression and ball milling experiments on **PN1**. These studies have been added to the revised manuscript in the section "bulk mechanochemical activation". While the extrusion and compression showed minimal ring opening, ball milling successfully activated the mechanophore, and the subsequent depolymerization showed 6 mol% DHF (compared to olefins in the original **M1** monomer). Degradation of the crosslinker allowed us to characterize the extent of ring opening (17%) using ^1H NMR, demonstrating a more economical approach of mechanochemical activation and degradation.

Reviewer: 2

Comments:

In the present paper, Hsu et al. depict a new strategy to attain poly(2,5-dihydrofuran) mechanochemically from an unsaturated polyether cyclic alkene with

a cyclobutene-fused as the mechanically active handle. This paper is extremely well written and well thought out. It builds upon some of the state-of-the-art on locked degradability. This reviewer has very little to argue from a technical perspective. Where the paper falls a bit short of the lofty goals it sets out from the onset is the usefulness of such material. The authors do not characterize either the processability (which may trigger activation) or the mechanical properties of the materials they synthesize? Would the ring opening occur under extrusion conditions or under load?

We thank the reviewer for the comment on the processing and mechanical properties, and we completely agree that these aspects are key to the usefulness of the material. We have added extrusion and compression studies in the revised manuscript. Specifically, a blend of **P1** and PEG was extruded on a twin screw extruder. ¹HNMR spectroscopy of the extruded material and subsequent depolymerization experiment revealed little ring opening of **P1**. Since **P1** is a polymer melt at room temperature ($T_g = -17\text{ °C}$), to evaluate its mechanical properties, a polymer network **PN1** was prepared by copolymerizing **M1** with a bis-cyclooctene crosslinker. Compressive testing of **PN1** showed Young's modulus of $1.03 \pm 0.19\text{ MPa}$. Little to no ring opening product was found in the extrusion product of **P1**/PEG blend or in the compression product of **PN1**. However, ball mill grinding of **PN1** gave an appreciable extent of ring opening (17%).

Reviewers' Comments:

Reviewer #1:

Remarks to the Author:

In their revision, the authors picked up on the suggestions and investigated the degradation of the mechanoresponsive latent low ceiling temperature polymer using extrusion and surprisingly they found little activation and degradation. However, the authors went far beyond what was requested and performed studies in crosslinked networks by compression and by trituration as well. In fact, the circumstance that the polymer does not degrade significantly during extrusion (manufacturing) and compression (cyclic loading in use) but by ball milling renders the results even more interesting. A very notable work and I can fully recommend its publication.

We thank the reviewers for their thoughtful comments and suggestions. Responses can be found below in blue text.

Reviewer #1

Comments:

In their revision, the authors picked up on the suggestions and investigated the degradation of the mechanoresponsive latent low ceiling temperature polymer using extrusion and surprisingly they found little activation and degradation. However, the authors went far beyond what was requested and performed studies in crosslinked networks by compression and by trituration as well. In fact, the circumstance that the polymer does not degrade significantly during extrusion (manufacturing) and compression (cyclic loading in use) but by ball milling renders the results even more interesting. A very notable work and I can fully recommend its publication.

We thank the reviewer for their very positive comments, and we completely agree that the stability during extrusion and compression but accessible degradation during ball milling makes the results more interesting.